# Predictive Modeling for Occupational Safety Outcomes and Days Away from Work Analysis in Mining Operations

**DOI:** 10.3390/ijerph17197054

**Published:** 2020-09-27

**Authors:** Anurag Yedla, Fatemeh Davoudi Kakhki, Ali Jannesari

**Affiliations:** 1Department of Computer Science, Iowa State University, Ames, IA 50014, USA; 2Department of Technology, College of Engineering, San Jose State University, San Jose, CA 95192, USA

**Keywords:** machine learning, word embedding, neural networks

## Abstract

Mining is known to be one of the most hazardous occupations in the world. Many serious accidents have occurred worldwide over the years in mining. Although there have been efforts to create a safer work environment for miners, the number of accidents occurring at the mining sites is still significant. Machine learning techniques and predictive analytics are becoming one of the leading resources to create safer work environments in the manufacturing and construction industries. These techniques are leveraged to generate actionable insights to improve decision-making. A large amount of mining safety-related data are available, and machine learning algorithms can be used to analyze the data. The use of machine learning techniques can significantly benefit the mining industry. Decision tree, random forest, and artificial neural networks were implemented to analyze the outcomes of mining accidents. These machine learning models were also used to predict days away from work. An accidents dataset provided by the Mine Safety and Health Administration was used to train the models. The models were trained separately on tabular data and narratives. The use of a synthetic data augmentation technique using word embedding was also investigated to tackle the data imbalance problem. Performance of all the models was compared with the performance of the traditional logistic regression model. The results show that models trained on narratives performed better than the models trained on structured/tabular data in predicting the outcome of the accident. The higher predictive power of the models trained on narratives led to the conclusion that the narratives have additional information relevant to the outcome of injury compared to the tabular entries. The models trained on tabular data had a lower mean squared error compared to the models trained on narratives while predicting the days away from work. The results highlight the importance of predictors, like shift start time, accident time, and mining experience in predicting the days away from work. It was found that the F1 score of all the underrepresented classes except one improved after the use of the data augmentation technique. This approach gave greater insight into the factors influencing the outcome of the accident and days away from work.

## 1. Introduction

Occupational injuries are a significant problem in the mining industry [1]. The Mine Safety and Health Administration (MSHA) reported that there were a total of 16,394 non-fatal lost time injuries in the US from 2015 to 2018 [2]. A total of 104 fatalities were reported in the US from 2015–2018 [2]. It is essential to analyze past mining injuries data to identify the factors leading to accidents and utilize them as predictors for future injuries [3].

Several researchers in mining safety and health have examined lost time from work in studies related to occupational safety. Among underground coal and metal/non-metal miners, both age and time away from work after an injury were observed to be directly proportional [4]. Onder used logistic regression to classify the accidents into two classes (greater than and less than three lost workdays). Maintenance personnel and workers were found to have the highest probability of exposure to accidents with greater than three lost workdays [5]. Handling materials is the most common type of injury resulting in lost-time injuries [6,7]. Coleman and colleagues discussed using lost workdays as a measure to compare the performance of safety programs in the absence of denominator (number of workers exposed) data [8]. Nouwrouzi and colleagues analyzed the top 56 annual and lifetime cited articles related to mining injuries [9]. Seventy-three per cent of the articles described certain factors as predictors of lost-time injuries. Every injury event has some causal factors associated with it. Injury outcomes can be predicted when the causes are known [10].

Statistical and machine learning techniques have been commonly used to analyze the importance of contributing factors toward an accident in the domain of occupation safety [10,11,12,13,14]. Classification trees, support vector machines, and neural networks are the most widely used machine learning models in this domain.

Matias and colleagues analyzed floor-level falls in the mining, construction, and services sectors using several machine learning techniques [11]. Bayesian networks, classification trees, support vector machines, and extreme learning machines were used in their approach. Bayesian networks were found to be the best all-round technique. Rivas and colleagues modelled incidents and accidents in two companies in the mining and construction sectors [15]. They also reported a similar result where Bayesian networks and classification trees outperformed logistic regression. The dataset comprised of survey results and information obtained from interviews. A similar superior performance of machine learning techniques was also reported in a study done by Tixier and colleagues on construction injury data [12]. Random forest and Stochastic Gradient Tree Boosting were used to predict three safety outcomes, namely, injury type, energy type, and body part.

Many studies in occupational safety domain use structured data to analyze the accidents [13,16]. Use of textual reports to predict the safety outcomes has been minimal. Marucci-Wellman and colleagues used Naive Bayes, Single word, and Bi-gram models, support vector machine, and logistic regression for classifying the event leading to injury using injury narratives of a workers compensation dataset [16]. Sarkar and colleagues used topic modelling on the injury text data to extract topics or classes [10]. A topic or class is a cluster of similar words. Davoudi Kakhki and colleagues used cluster modeling to identify high-risk groups of occupational incidents with severe injuries [17]. Previous studies on occupational safety outcomes have shown the superior performance of machine learning techniques, such as classification trees, support vector machines, and artificial neural networks (ANNs) [18,19].

The potential benefits of machine learning algorithms can not only be recognized by their capability to handle multi-dimensional and large amounts of data, but also from (i) their ability to generate actionable insights to improve decision-making, and (ii) their capacity to improve over time with exposure to more data [10,12]. Due to the advantages of machine learning techniques, they have been successfully used in many fields, such as e-commerce, healthcare, and banking [20,21,22]. Studies on occupational safety studies have also used machine learning techniques to analyze and generate actionable insights [12,15,18,23]. Similarly, the use of machine learning techniques can significantly benefit the mining industry. Large amounts of safety-related data related to mining is available, and humans cannot review all the data. The machine learning techniques can be leveraged to analyze safety data and provide actionable insights. However, the use of machine learning methods in the mining industry has been minimal [23,24]. Use of text narratives in the predictive analysis is infrequent [16]. Similarly, not many studies use days away from work as an outcome while analyzing mining injuries using machine learning techniques. Modelling the outcome of the injury and days away from work is vital to identify the factors leading to these outcomes. Days away from work (DAFW) is also an indicator of the severity of the injury. The purpose of this study is to identify the potential of text narratives to predict the outcome of the injury and days away from work. We used logistic regression, decision trees, random forests, and ANNs to find answers to the following questions: (1) Do text narratives have enough information to predict the outcome of the injury compared to the tabular data, and (2) can text narratives be used to predict days away from work? Decision trees, random forest, and ANN were selected based on the superior performance of these models in the studies related to occupational safety [10,12,13,15,16]. The performance of these models were compared to the performance of logistic regression.

## 2. Materials and Methods

### 2.1. Data

The MSHA accident injuries dataset was used in this study. The dataset is publicly available and was obtained from the United States Department of Labor website. The dataset contains information about the accidents reported by mine operators and contractors in the US between 2000 and 2018 [2].

### 2.2. Logistic Regression

Logistic regression has been widely used to model the odds of an outcome in the analysis of categorical data. It was first developed by Dr. Cox in 1960 [25]. Logistic regression is used when the target variable (dependent) is categorical. Linear regression is not suitable for classification problems as the output of the linear regression model is continuous and not bounded. On the contrary, the output of the logistic regression model is always limited to values between 0 and 1. The logistic regression equation can be written as shown in Equation (Equation 1),
(1)$$logit\left(p\right)=ln\left(\frac{p}{1-p}\right)=\beta_{0}+\beta_{1}x_{1},$$
where the left-hand side (LHS) of the equation is the natural logarithm of the odds ratio, and right-hand side (RHS) is a linear function of the independent variables. The equation can be rewritten to find the estimated probability, as shown in Equation (Equation 2).
(2)$$p=\frac{1}{1+e^{-(\beta_{0}+\beta_{1}x_{1})}}$$

Logistic regression can handle any number of numerical or categorical dependent variables, as shown in Equation (Equation 3).
(3)$$p=\frac{1}{1+e^{-(\beta_{0}+\beta_{1}x_{1}+\beta_{2}x_{2}+\ldots +\beta_{n}x_{n})}}$$

The regression coefficients for logistic regression are calculated using maximum likelihood estimation (MLE) [26]. Logistic regression is easy to train and interpret. However, logistic regression is useful when working with a linearly separable target class [27]. Logistic regression has been widely used in studies related to occupational safety [5,14,15,16].

### 2.3. Decision Tree

A decision tree is a flowchart-like structure, where each leaf node represents a class label, each non-leaf node (internal node) represents a test on a dependent variable (attribute), and each branch represents an outcome of the test [28]. Decision trees can be converted to classification rules by tracing the path from the root node to each leaf node. The decision for a tuple X, with an unknown class label, is made by testing the attribute values of the tuple against the decision tree. Domain knowledge and parameter setting are not required to construct decision trees [28]. These properties make decision trees a popular choice for exploratory knowledge discovery. ID3 (Iterative Dichotomiser), C4.5, and Classification and Regression Trees (CART) are some of the famous algorithms used to construct decision trees. Attribute selection is an essential part of constructing the decision tree. At every level of the tree, the attributes that best partition the tuples into distinct classes are chosen [29]. Some techniques, such as tree pruning, are used to improve classification accuracy on test data or unseen data [30]. Decision trees are very fast and easy to interpret. However, decision trees tend to overfit on training data, and often a single tree is not enough to produce good results [31]. Due to easy interpretability, decision trees are used quite often in injury analysis studies [11,15].

### 2.4. Random Forest

Random forest is an ensemble model where each of the classifiers in the ensemble is a decision tree classifier [28]. Ensemble learning is the method of combining several models to make the final prediction [32]. It helps in reducing variance, bias, and improving performance [33]. Bagging is used to reduce the variance, and boosting is used to reduce bias. Random forest uses bagging as the ensemble method and decision trees as individual models. Random subsets of the dataset are created, and by using the subsets, decision trees are created. Each decision tree is built by selecting random attributes at each node to determine the split. All the decision trees participate in a majority vote, and the most popular vote is chosen as the target class or label. Random forest reduces overfitting as it averages over the independent trees [28]. They are more robust to errors and outliers [28]. They can also be used as a feature selection tool. Since the random forest is a collection of decision trees, they are computationally expensive and are difficult to interpret [31]. Random forests were used in several studies related to injury analysis [34,35].

### 2.5. Artificial Neural Network

Like the random forest, an ANN is also used for classification and regression purposes. The ANN model is inspired by the functioning of the human brain. It is a network of artificial neurons arranged in three kinds of layers—an input layer, one or more hidden layers, and an output layer [28,36]. The neurons in the network are also referred to as units. Units in each layer are connected to all the units in the adjacent layers. The number of features or attributes of the tuple dictates the number of units in the input layer. Similarly, the number of units in the output layer depends on the number of class labels. During the training process, information is fed into the network via the input units, which passes through the units in hidden layers and arrives at the output units. Each unit computes the weighted average of its input, and the sum is passed through a non-linear function [37]. The unit fires when the sum is higher than a specific threshold value, and it triggers the units it is connected to in the next layer. An element of feedback is involved in the learning process. The output produced by the network is compared with the actual output it was supposed to produce, and the difference is used to modify the weights associated with the connections between the units. This learning algorithm is known as back propagation [38,39]. ANNs can detect complex nonlinear relationships between explanatory and outcome variables. However, they are computationally expensive, and it is hard to interpret the trained model [28,40]. According to previous studies, ANN was successfully used in construction injury prediction [10].

### 2.6. Measures

#### 2.6.1. Explanatory Variables

A total of 50 characteristics of the injury were present in the dataset. Columns related to dates and codes, such as mine ID, controller ID, operator ID, contractor ID, subunit code, calendar year, calendar quarter, fiscal year, fiscal quarter, degree injury code, fips state code, ug location code, ug mining method code, mining equipment code, equipment manufacturer code, equipment model number, classification code, accident type code, occupation code, activity code, injury source code, nature of injury code, injured body part code, schedule charge, and closed document number were removed. Columns like mining equipment, the underground mining method, and underground mining location were removed, as more than half of the entries had a value of “no value found”. Fifteen variables remained after removing all the unrelated columns to this study. Categorical variables in this dataset included subunit, classification, accident type, occupation, activity, injury source, nature of the injury, injured body part, and narrative. The nature of the injury identifies the injury with regard to the principal physical characteristics, and has 38 categories. For example, burn, cut, sprain, and hearing loss are some of the categories. Occupation has 198 categories (e.g., welder, rock driller, and sandfiller). The injured body part variable has 47 categories. The upper extremities, back, and forearm are examples of some specific injured body parts. Activity refers to the specific activity that the victim was performing at the time of the accident. There are 98 categories of this variable. For example, some of the activities are grinding, handling supplies, and machine maintenance. Accident type identifies the event which directly resulted in the reported injury/accident. Forty-three kinds of accident types are present in this dataset. Some of the accident types are struck by flying object, fall to the walkway, and fall down the stairs. The sub-unit variable has five categories, each referring to the location within a mine where the accident/injury/illness occurred. For example, the specific subunits are underground, the surface at underground, strip/quarry/open pit, auger, and culm bank/refuse pile. Classification refers to the circumstances which contributed most directly to the resulting accident. Some of the values for classification are entrapment, stepping, or kneeling on object, and striking or bumping. The classification variable has 28 categories. Injury source is the object, substance, exposure, or bodily motion that directly produced or inflicted the injury. Injury source has 127 categories. Some examples of injury sources are ladders, steel rail, and caving rock. The variable of coal/metal industry was initially a categorical variable, but it was changed to a numerical variable by substituting the value “C” representing coal to 0 and “M” representing metal industry to 1. Accident time and shift start time are the two time-related variables included in this dataset. Numerical variables included are total experience (total mining experience), mine experience (experience at a specific mine), and job experience (experience in the job title). The narrative column is restricted to a specific character limit, because of which the narratives are not very long. Some examples of narratives are: (i) “Employee was lifting dewatering press cloths when he heard a ’pop’ in his left elbow area”; and (ii) “Plant Foreman was doing his routine shift walk around; on his way down through the rear stairs from the control room to the clinker storage silos, one of the stair treads broke loose, causing the employee to lose his balance; his right leg went through the opening and he fell frontward, hitting his right knee on the next tread”. Employee suffered a muscle sprain and mild knee contusion.

#### 2.6.2. Outcome Variables

Degree of injury and days lost are the two outcome variables used in this study. Different values in the degree of injury column were accident only, fatality, permanent total or permanent partial disability, days away from work only, days away from work and restricted activity, days restricted activity only, no days away from work, no restrictions, occupational illness not degree 1–6, injuries due to natural causes, injuries involving non-employees, and all other cases (incl. first aid). All the rows with values as “Injuries involving non-employees” were removed since the study concentrated on the injuries incurred by the employees. Categories of the degree of injury variable are shown in Table 1.

### 2.7. Data Pre-Processing

Data preprocessing is the most critical step in the machine learning pipeline. Preprocessing, if done well, could boost the model performance [41]. All the rows containing empty columns were removed. All the stop words (i.e., commonly used words, such as “a” or “the”) were removed from the injury narratives. Stemming was performed on all the words in the narratives. Stemming is the process of reducing a word to its root form, that is, reducing words such as “laughing” or “laughed” to “laugh”. Most of the variables in the fixed field entries were categorical (a variable that can take on a limited number of values), and some of them had high cardinality. Categorical variables with high cardinality are often challenging as an input for machine learning models, such as ANNs [42]. While there are many techniques to deal with such variables, the following technique was used to encode the categorical variables.

#### 2.7.1. Categorical Encoding Using Target Statistics

One-hot encoding is one of the most common encoding techniques. In one-hot encoding, new binary columns are created, which indicate the presence of each possible value from the original data. Figure 1 shows the one-hot encoding of a categorical variable, color, which has three categories. While one-hot encoding is the most popular encoding technique, it has certain limitations [43]. One hot encoding generates many binary variables when the cardinality of the categorical variable is high, that is, when the categorical variable contains many distinct values. This type of encoding leads to an increase in the number of features. For high cardinality categorical features, encoding using target variable statistics can be used [43]. Let Y be a multi-valued categorical target variable, where Y ∈ $Y_{1}$, $Y_{2}$…$Y_{m}$. For each possible value $Y_{j}$ of the target variable, a derived variable $X_{j}$ is created in substitution of the original high cardinality categorical independent variable X. Each derived variable $X_{j}$ will represent an estimate of $P(\frac{\left(Y=Y_{j}\right)}{\left(X=X_{i}\right)})$ using the formula shown in Equation (Equation 4),
(4)$$S_{i}=\lambda \left(n_{i}\right)\frac{n_{iY}}{n_{i}}+\left(1-\lambda \left(n_{i}\right)\right)*\frac{n_{Y}}{n_{TR}},$$
where $n_{TR}$ is the total number of records, $n_{iY}$ is the number of records belonging to class Y for a particular value in the column, and $n_{Y}$ is the number of records belonging to class Y. Since the sum of the probabilities is 1, creating k derived variables is redundant. Thus, we introduce only k − 1 derived variables and drop any one of the $X_{j}$. Generally, a function with one or more parameters is chosen as λ(n). The parameters of λ(n) can be adjusted depending on the data. We chose λ(n) as a parameter function shown in Equation (Equation 5),
(5)$$\lambda \left(n\right)=\frac{n}{n+m},$$
where m is a constant. Figure 2 shows an example of encoding the variable color using target statistics where the target variable is fruit.

#### 2.7.2. Word Embedding

Word embeddings are used for vector representation of words in natural language processing (NLP) [44]. When words are treated as atomic units, the notion of similarity cannot be expressed, whereas when words are represented as vectors, cosine distance can be calculated between two words to check the similarity between them. In NLP tasks, the performance of learning algorithms is boosted when words are represented in a vector space [45]. Word2vec is a word embedding technique used to learn high-quality vector representations of words [44,46].

Word2vec trains a neural network with one hidden layer with words against the neighboring words within a large corpus of text. There are two kinds of learning models for Word2vec, and both do the same thing but in different ways. One learning model is called the continuous bag of words (CBOW), and the other is called the skip gram model [44]. CBOW uses context to predict the target word, whereas the skip gram model uses the target word to predict context.

### 2.8. Representation of Narratives

We trained the Word2vec model with the narratives in the MSHA dataset. All the narratives were divided into tokens (words), as shown in Figure 3. Then, using the trained Word2vec model, each word was represented as a vector of length 300. The vector representation of each word was multiplied with the term frequency and inverse document frequency (TF-IDF) score, as shown in Figure 4. Term frequency is the ratio of the frequency of the term in the narrative and the total number of terms in the narrative. Inverse document frequency is the logarithm of the number of the narratives in the corpus divided by the number of narratives where the specific term appears. The TD-IDF score is the product of term frequency and inverse document frequency. Then, vector representations of the words in the narrative were added and averaged. The resulting vector is the vector representation of the narrative with 300 components. Figure 4 shows the process of converting narratives to vectors.

### 2.9. Data Augmentation

Imbalance in the target classes can often lead to poor performance of the predictive models [12]. The dataset used in this study is highly imbalanced. We used synthetic data augmentation to tackle the data imbalance problem. We also used word embeddings to generate fake narratives [47]. First, we created nine different Word2vec models, one for each target class. Then, we randomly chose six words that will be replaced in each narrative in the training set. We replaced each of the six words with the top three closest words. The top three closest words were determined using the trained Word2vec models for the respective classes. This way, we could generate 18 narratives from one narrative. We did not replace words in the narrative if they were shorter than six words.

### 2.10. Performance Metrics

Various evaluation metrics are available to understand the performance of the models. Figure 5 shows a simple confusion matrix. A confusion matrix is a technique to evaluate the performance of a classification algorithm. Each row in the confusion matrix represents an actual class, and each column represents a predicted class. Performance metrics used for classification and regression tasks in this study are discussed in this section.

#### 2.10.1. Accuracy

Accuracy is the ratio of the number of correct predictions and the total number of predictions. The formula for accuracy is shown in Equation (Equation 6).
(6)$$accuracy=\frac{true\;positive}{total\;number\;of\;samples}$$

#### 2.10.2. F1 Score

Accuracy is not a good measure when the target variable classes in the dataset are imbalanced [48]. A model which predicts the target class as the majority class for every input can achieve a high accuracy score. We used the F1 score as a performance measure in this study. The F1 score is the harmonic mean of precision and recall. Precision is the proportion of positive identifications that are actually correct, and recall is the percentage of the total number of positive instances that are correctly classified. Both precision and recall are essential for this study, so we used the F1 score as a metric that combines precision and recall. The formula for precision and recall are given in Equations (Equation 7) and (Equation 8). We calculated the F1 score for each target class, and their average was weighted by support (number of samples).
(7)$$precision=\frac{true\;positive}{true\;positive+false\;positive}$$
(8)$$recall=\frac{true\;positive}{true\;positive+false\;negative}$$

Formula for calculating F1 score is given in Equation (Equation 9).
(9)$$F1score=2*\frac{precision*recall}{precision+recall}$$

### 2.11. Mean Squared Error (MSE)

For regression models, MSE is widely used as a performance metric. It represents the average of the squares of the differences between predicted values and observed values. MSE was calculated as shown in Equation (Equation 10).
(10)$$MSE=\frac{1}{n}\sum_{j=1}^{n}{\left(y_{act}-y_{pred)}\right)}^{2}$$

### 2.12. Root Mean Square Error (RMSE)

RMSE is the square root of MSE. RMSE was calculated as shown in Equation (Equation 11).
(11)$$RMSE=\sqrt{\frac{1}{n}\sum_{j=1}^{n}{\left(y_{act}-y_{pred)}\right)}^{2}}$$

In the Equation (Equation 11), $y_{act}$ represents the actual value and $y_{pred}$ represents the predicted value.

### 2.13. Predicting Outcome of the Injury

Two kinds of data were used in this experiment—fixed field entries, and narratives. The target variable was the degree of injury. After removing all the target classes with entries less than 1% of the dataset, three classes remained. The dataset consisting of 110,996 entries was split into training (70%) and testing (30%) sets. Stratified random sampling was used to split the dataset [49].

#### 2.13.1. Fixed Field Entries

Logistic regression, decision tree, random forest, and ANN were used. For decision tree, the Gini index was used as the attribute selection measure. For random forest, the number of decision trees in the forest was chosen as 30, Gini index was used as the attribute selection measure. The parameters used for ANN were as follows: two hidden layers, rectified linear units as the activation function for hidden layers, softmax as the activation function for output layer, a learning rate of 0.001, and drop out rate of 0.3. In the initial model we combined accident time and shift start time into one variable by computing the difference in hours between accident time and shift start time. Combining accident time and shift start time reduced the overall accuracy of the models. In the final model we used both accident time and shift start time.

#### 2.13.2. Narratives

The input to the models was the vector representation of the injury narratives, which was computed as shown in Section 2.8. The parameters for the decision tree and random forest were the same as those used for fixed field entries. ANN was trained on unbalanced (nine target classes) and augmented datasets. Table 1 shows the number of narratives added to each imbalanced class in a training set using synthetic augmentation. The test dataset remained the same for ANN when trained on the unbalanced and augmented dataset. The parameters used for ANN were the same as used in fixed field entries, except for the number of neurons in the input layer.

### 2.14. Predicting Days Away from Work

Two kinds of data were used in this experiment—fixed field entries, and injury narratives. The target variable was the number of days lost due to the injury. The dataset consisting of 79,457 records was split into training (70%) and testing (30%) using stratified random sampling. All the records with no days lost were removed.

#### 2.14.1. Fixed Field Entries

The target variable was days away from work. Random forest and ANN were used. For random forest, the number of decision trees in the forest was chosen as 30. Mean Squared Error (MSE) was used as the function to measure the quality of a split. The parameters used for ANN were as follows: two hidden layers, rectified linear units as the activation function for hidden layers, the learning rate of 0.001, and drop out rate of 0.3. Softplus activation function was used as the activation function for the output layer to prevent the model from predicting negative values. MSE was used as the performance metric. In the initial model we combined accident time and shift start time into one variable by computing the difference in hours between accident time and shift start time. This reduced the overall accuracy of the models. In the final model we used both accident time and shift start time.

#### 2.14.2. Narratives

The input to the models was the vector representation of the narratives, which is computed, as shown in Section 2.8. All the parameters for random forest and ANN were similar to the parameters used in the fixed fields entries section of predicting days away from work, except the number of neurons in the input layer for ANN. Keras and Sklearn (machine learning libraries in python) were used to build all the models.

## 3. Results

In this section, we show and compare the performance of all the models in predicting injury outcome and days away from work. The results are in two parts. In the first part, we show and compare the performance of logistic regression, decision tree, random forest, and ANN (with fixed field and injury narratives as input) in predicting the injury outcome. In the second part, we show and compare the performance of logistic regression, decision tree, random forest, and ANN (with fixed field and injury narratives as input) in predicting days away from work.

### 3.1. Injury Outcome

Logistic regression, decision tree, random forest, and ANN were used to predict the injury outcome. We used two kinds of inputs—fixed field entries, and injury narratives. Table 2 shows the overall accuracy and F1 score of the models with fixed field entries as input. All the models had decent performance, except the decision tree. ANN had the best overall accuracy of 78%. Logistic regression and random forest had an accuracy of 67% and 66%, respectively. ANN was also the best model in terms of F1 score. ANN had an F1 score of 0.67. Logistic regression and random forest had an F1 score of 0.64 and 0.65, respectively. Logistic regression had an F1 score comparable to random forest. Overall, ANN performed better than all other models. Decision tree had the least accuracy (58%) and F1 score (0.58).

Table 3 shows the F1 score and overall accuracy of random forest and ANN trained on imbalanced injury narratives. Random forest had the highest F1 score (0.94) and accuracy (94%) among both the models. Figure 6 shows the confusion matrix of random forest trained on the injury narratives. Figure 7 shows the F1 score of ANN on unbalanced and balanced (using synthetic augmentation) datasets. The F1 score of the underrepresented classes 1, 5, 8, and 9 improved after augmentation. The F1 score for the underrepresented class 6 decreased after augmentation. Data augmentation was not performed for classes 2, 3, 4, and 7. The F1 score for classes 2, 3, and 7 decreased. Class 4 did not have any impact on the F1 score. The overall F1 score of ANN on the unbalanced dataset was 0.60. After augmentation, the F1 score decreased to 0.58.

### 3.2. Days Away from Work

Random forest and ANN were used to predict DAFW. RMSE was used as the metric to compare the performance. Similar to injury outcome prediction, we used two kinds of inputs—fixed field entries, and injury narratives. The standard deviation of DAFW variable in the dataset was 75.02. Table 4 shows the MSE and RMSE for all the models with fixed field entries as input. ANN had the best performance compared to others. RMSE for ANN was 0.62. Random forest had a RMSE value of 3.82. Table 4 also shows the RMSE for ANN with injury narratives as input. Overall, ANN with fixed fields entries as input performed better than all other models.

### 3.3. Feature Importance

Due to the unstructured nature of text narratives, it was not possible to identify the most helpful features to predict the target class. Thus, we used the ANN trained on fixed fields to analyze the feature importance. ANN is trained each time by removing one independent variable (feature) from the dataset. We computed the difference between the overall F1 score of the ANN trained on the complete dataset and the ANN trained on the dataset with one missing feature. This difference represents the feature importance. Table 5 lists the features in descending order of feature importance.

## 4. Discussion

This study used supervised machine learning techniques, such as logistic regression, decision tree, random forest, and ANN to predict injury outcome and days away from work in mining operations. Fixed field entries (structured data) and injury narratives (unstructured data) were used to train the models. The experiments done in this study show that random forest trained on the vector representation of injury narratives performed better than all other models. The high accuracy and an F1 score of random forest even when there exists a class imbalance shows the effectiveness of ensemble learning methods. The most plausible conclusion for the superior performance of random forest on narratives is the information present in the narrative, which is not present in the fixed field entries. One example would be the narrative, “Employee was grinding off metal that had been cut on flop gate. A piece of metal must have got under employee’s safety shield and safety glasses, causing an abrasion to eye. Employee did not notice discomfort until employee got home”. The model trained on fixed field entries classified this incident as “Days restricted activity only”. The tabular data does not have any mention of safety glasses or safety shield that the employee was wearing. However, the model trained on narratives has information about the safety equipment. This could be one of the reasons why the model trained on narratives classified this incident correctly, which is “No days away from work, No restricted activity”. ANN performed relatively better than other models when the input was fixed field entries. However, underrepresented classes were removed from the dataset when fixed field entries were used. This study forms a foundation for the future research in utilizing text narratives in the predictive analysis of injury outcomes.

The mining industry exceeds many industries in terms of workplace injuries and fatalities [50,51,52]. It is, therefore, essential to study the characteristics of the mining injuries in order to find the factors leading to an injury. Once the factors associated with the injuries are identified, safety programs can be designed to address those issues. This study used ANN trained on fixed field entries to find the feature importance. According to Table 5, the nature of the injury is the most influential feature in the dataset. The nature of the injury was also found to be a significant variable in the prediction of injury severity level in the agribusiness industry [13]. Since our focus was on injuries causing lost days of work, we analyzed the nature of injury variables for the injuries resulting in DAFW. The highest number of nature of injuries resulting in DAFW were sprain, disc rupture, fracture, cut, laceration, and bruise. The second most influential variable was an injured body part. The injuries to the back, spine, Scord, and tailbone were among the highest to result in DAFW. Occupation was also one of the essential features. An injury to the workers having the following occupations: Maintenance man, Mechanic, Repair/Serviceman, Boilermaker, Fueler, Tire tech, and Field Service tech had the highest probability to result in DAFW class. Maintenance personnel and workers were also found to have the highest risk of occupational injuries among opencast coal mine workers [5]. Job experience was among the top five important variables which were important in predicting the outcome of the injury. It is interesting to note that job experience had much higher feature importance compared to mine experience and total experience when predicting the outcome of the injury. Margolis analyzed how age and experience were related to days away from work in underground coal mining injuries. It was found that the total mining experience has an influence on the severity of the injury [4]. Mine experience and job experience were found to have no effect on the severity of the injury. However, it needs to be noted that the dependent variable was the number of days away from work, whereas in our study, the dependent variable was the outcome of the injury.

Given the high predictive power of the model, the above variables are significant in predicting the outcome of the injury. The nature of the injury is the most important predictor, and sprain, disc rupture, and fracture result in the most days away from work. Safety programs can be designed specifically to reduce accidents of this nature. These safety programs should also concentrate on other influential variables found in this study, such as injured body parts and occupation. Since job experience was found to be more important in predicting the outcome of the injury than mine experience and job experience, emphasis should be given to job-related safety.

Although the model with the best performance cannot be used to analyze feature importance, it can certainly help to answer questions such as, “if this kind of injury were to happen, what would it result in?”, and, “What if a different body part was injured rather than the body part mentioned in the narrative?” Answers to such questions would help safety managers to plan for accidents that could occur in the future.

The data augmentation using word embedding increased the F1 score of ANN for unbalanced classes, except for one class. Although the overall F1 score of the model decreased from 0.60 to 0.58, the decrease in the performance was not very significant. One of the reasons for the decrease in the overall accuracy could be the way the words to be replaced were chosen. Since they were chosen randomly, the target class of the fake narrative could have changed from the target class of the original narrative. Having longer narratives would have helped in generating more accurate synthetic narratives generation.

Models trained on fixed field entries performed better than the models trained on narratives when predicting the DAFW. ANN trained on fixed field entries had the least MSE. It is interesting to note that the only information missing from the narratives that is present in the fixed field entries is the shift start time, accident time, and the experience of the miner. The presence of the above variables in the fixed field entries could have helped the model in predicting the DAFW better. Accurately predicting DAFW could help the supervisors managing the workforce to plan for replacements when an injury occurs. DAFW is also an indicator of the severity of the injury. These models are not a replacement to an expert in safety; instead, they are tools to help safety experts to act proactively to reduce workplace injuries.

## 5. Conclusions

We explored a new research problem of predicting the outcome of the injury and the number of days away from work in the mining industry using machine learning models. Target-based statistics were used to encode categorical variables. This technique helped to tackle the problem of high cardinality categorical variables. Random forest trained on injury narratives performed better than all the models. The high predictive power of the model trained on narratives suggests that the narratives contain additional important information compared to the fixed field entries. The synthetic data augmentation with word embedding was used to tackle the data imbalance problem. This technique improved the F1 score of ANN for the underrepresented classes. However, the overall accuracy and F1 score of the model decreased after augmentation. There is a lot of unstructured data available compared to the structured data, and the results of this study show that using unstructured data, such as text narratives, could be useful in understanding the injuries better. This study shows that there is a potential for using NLP and text analytics in this field.

Regarding the application of predictive modeling in occupational injury analysis, this study not only confirms the findings from previous work on the effectiveness of data mining techniques in analyzing occupational incidents in the mining industry [53], but also adds new methods for dealing with limited data, and yet extracting useful practical information for improving safety of mining operations. However, there are some limitations in this work. There are no weather-related features in the data, and it makes it hard to analyze if the severity of days-away-from-work classes could be impacted by weather conditions. In addition, the data lacks information about demographics of the injured workers, and thus, does not explore the role of age and experience of the worker on the days-away-from-work severity classes. Furthermore, some studies have analyzed and compared the risk of occupational incidents in mining based on workers’ genders [54]. Therefore, another limitation to this study is a lack of probabilistic risk analysis based on the injured worker’s gender. Future direction of this research includes exploring new data collection methods to improve the quality and features of mining occupational incidents and further building models for estimating probability of each days-away-from-work class based on the features of the incident. Deep learning techniques, such as convolutional neural networks (CNNs) and recurrent neural networks, have shown promising results in text classification [55,56]. Future studies can expand on this study by using such deep learning models. Generative adversarial networks (GANs) have been used in recent works for text generation [57]. The use of GANs to tackle data imbalance problems in the domain of occupational safety could be explored.

## Figures and Tables

**Figure 1 ijerph-17-07054-f001:**
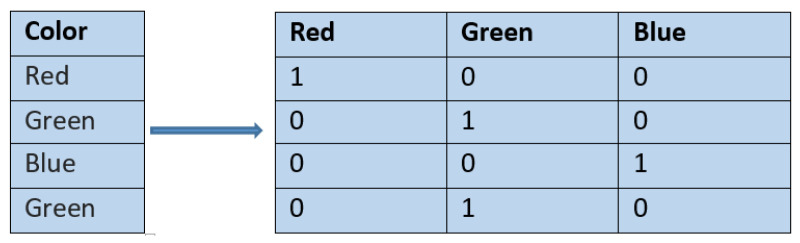
An example of one-hot encoding.

**Figure 2 ijerph-17-07054-f002:**
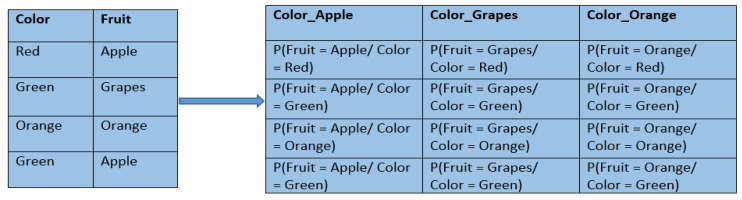
An example of encoding based on target statistics.

**Figure 3 ijerph-17-07054-f003:**
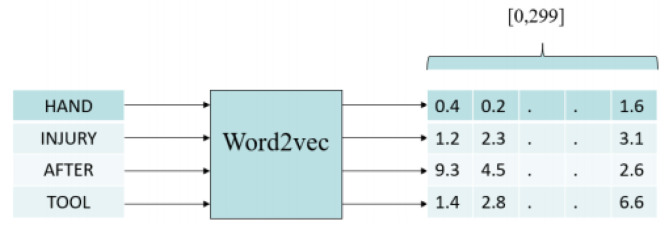
Converting each word to a vector of length 300.

**Figure 4 ijerph-17-07054-f004:**
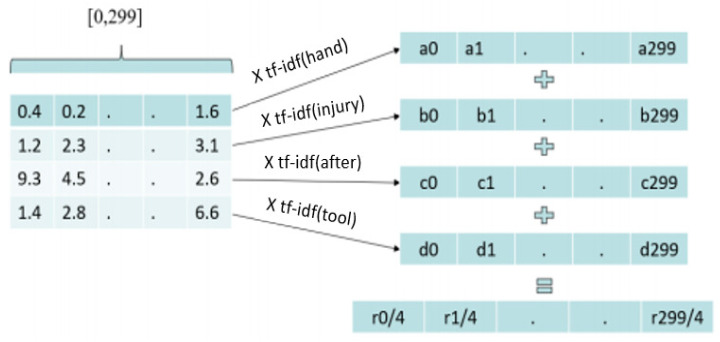
Vector representation of narratives.

**Figure 5 ijerph-17-07054-f005:**
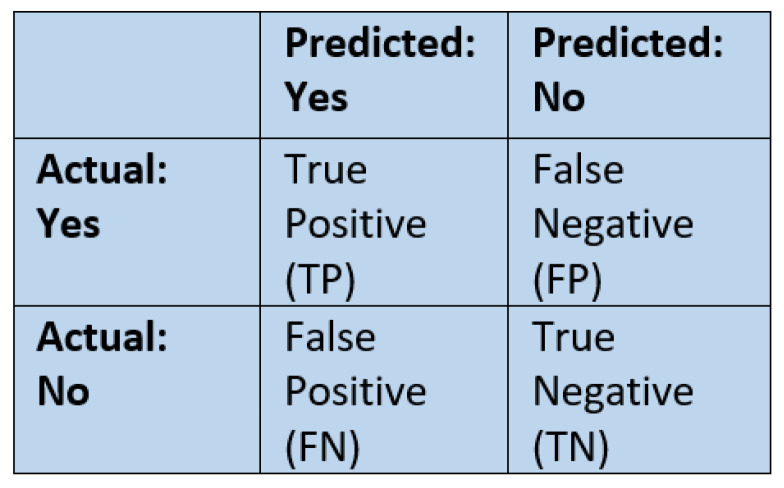
A simple confusion matrix.

**Figure 6 ijerph-17-07054-f006:**
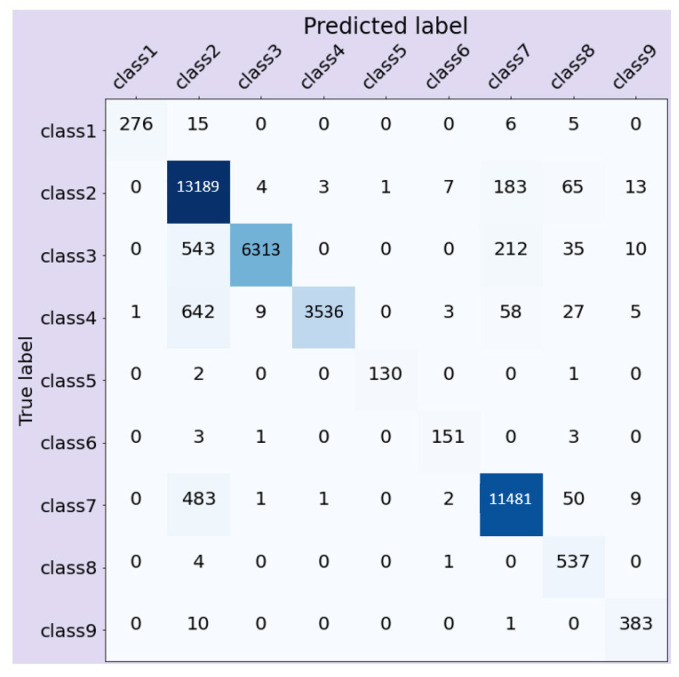
Confusion matrix for random forest trained on injury narratives.

**Figure 7 ijerph-17-07054-f007:**
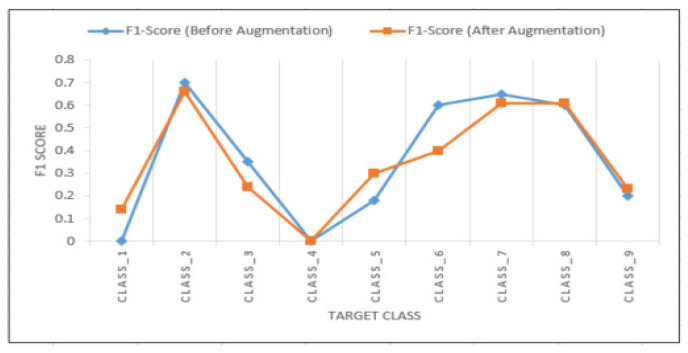
F1 score of artificial neural networks on unbalanced and augmented narratives.

**Table 1 ijerph-17-07054-t001:** Number of records in each target class before and after synthetic augmentation.

Target Class	Count before Augmentation	Count after Augmentation
Class1: All Other Cases (Including 1st Aid)	676	7564
Class2: Days Away From Work Only	31,653	31,653
Class3: Days Restricted Activity Only	16,633	16,633
Class4: Days Away From Work & Restricted Activity	10,025	10,025
Class5: Fatality	336	3842
Class6: Injuries due to Natural Causes	444	2785
Class7: No Days Away From Work, No Restricted Activity	27,627	27,627
Class8: Occupational Illness not DEG 1–6	1346	9676
Class9: Permanent Total or Permanent Partial Disability	895	12,796

**Table 2 ijerph-17-07054-t002:** Accuracy and F1 score for all the models (fixed field entries).

Model	F1 Score	Accuracy
Logistic regression	0.64	67%
Decision Tree	0.58	58%
Random Forest	0.66	66%
Artificial Neural Network	0.67	78%

**Table 3 ijerph-17-07054-t003:** Accuracy and F1 score for all the models (imbalanced narratives).

Model	F1 Score	Accuracy
Random Forest	0.93	93%
Artificial Neural Network	0.60	92%

**Table 4 ijerph-17-07054-t004:** MSE and RMSE for all the models.

Model	Input	MSE	RMSE
Random forest	Fixed Field Entries	14.65	3.82
	Injury Narratives	1502.61	38.76
Artificial neural network	Fixed Field Entries	0.38	0.62
	Injury Narratives	5944.74	77.10

**Table 5 ijerph-17-07054-t005:** Dependent variables and their description in descending order of their importance.

Feature	Description
Nature of Injury	Identifies the injury in terms of its principal physical characteristics.
Injured body part	Identifies the body part affected by an injury.
Occupation	Occupation of the accident victim’s regular job title.
Coal or Metal	Identifies if the accident occurred at a Coal or Metal/Non-Metal mine.
Job Experience	Experience in the job title of the person affected calculated in the decimal year.
Hours	Time difference between accident time and shift begin time in hours.
Injury Source	Identifies the object, substances, exposure or bodily motion which directly produced or inflicted the injury.
Classification	Identifies the circumstances which contributed most directly to the resulting accident.
Activity	Specific activity the accident victim was performing at the time of the incident.
Accident type	Identifies the event which directly resulted in the injury/accident.
Sub-unit	The Sub-unit of the mining site where the accident occurred.
Mine experience	Total experience at a specific mine of the person affected calculated in decimal years.
Total experience	Total mining experience of the person affected calculated in decimal years.

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
