# Peer review of "Predictive Modeling for Occupational Safety Outcomes and Days Away from Work Analysis in Mining Operations"

_ijerph, 2020, doi:10.3390/ijerph17197054_

Round 1

Reviewer 1 Report

In the background, it would be helpful to provide some information on the pros and cons of the different machine learning techniques, in addition to telling the reader what these techniques are.

In the study's justification, the authors should provide a brief argument on the potential benefits of machine learning methods in the mining sector. Simply saying their use has been minimal is not enough.

The discussion section could be restructured to improve flow and coherence. The authors start off rather abruptly. Consider starting by reminding your reader the purpose of the study. If you choose this route, you would have to move the first few sentences in the Conclusion section to the discussion section.

Reviewer 2 Report

With this work machine learning models have been used to predict days away from work in mining industry. Performance of all the models (decision tree, random forest and neural networks) has been compared with the performance of traditional logistic regression model. Random forest trained on injury narratives performed better than all the models.

This is an interesting paper on which I have some relatively minor comments. I recommend that authors consider my comments in order to improve their manuscript:

Abstract: between lines 4 and 5 it is recommended to add a small paragraph that allows justification as well as a smoother transition to machine learning models.

Introduction section: (i) it is advisable to increase the justification of the use of machine learning methods; (ii) optimize the use of acronyms, especially with artificial neural networks (ANN) when it is used the first time.

Materials and Methods section: (i) the methodological framework for the use and comparison of machine learning methods should be clearer and more explicit; (ii) Figure 2 is very small and makes it difficult to read.

Results section: (i) why Class 2 has not been included on line 247? (ii) on line 260, “section a)” is not necessary.

Discussion section: In my opinion this section needs to improve a lot. (i) there is no clear differentiation between analysis and discussion of results. In relation to the discussion itself, it is necessary to use a greater number of references that improve this discussion. Currently, the authors have only used one reference ([11]); (ii) The limitations are very poor because they focus only on the dataset; (iii) the assertion of negligence exposed on the line 266 is highly controversial.

Conclusions section: (i) it is recommended to reduce the summary of lines 316-320 in order to give greater importance to the conclusions themselves; (ii) the proposed future works between lines 329-334 need a less forced and more justified exposure.

Reviewer 3 Report

This paper used decision tree, random forest and neural networks to analyze the outcomes of mining accidents and predict days away from work. Overall, the paper is of poor quality, reflected by the following issues.

  1. In the introduction part, the authors mentioned that many researchers had investigated the factors that result in lost-time from work in mining. I would expect there is a detailed elaboration on the factors to let readers know what has been done in the relevant literature or what were the related work. Also, the structure of the introduction is messy.
  2. The authors should give a brief literature review on logistic regression, decision trees, random forests and artificial neural What are the advantages of these techniques? Why are they selected for this study?
  3. The explanatory variables included subunit, classification, accident type, occupation, activity, injury source, nature of the injury, injured body part, narrative. However, the categories of these variables were not given. It is important to provide such information to readers to understand what these variables really are. This suggestion is still applicable to the outcome variables. Table 5 does not include all selected variables in the data analysis. The authors should ensure all selected variables are included in Table 5.
  4. The results of data analysis only focused on the performance of predicting the injury outcome. The important factors that can be used to predict the injury outcome were not identified and discussed. I would expect the authors can provide such results and have a detailed discussion on it.
  5. In the discussion, it only focused on the statistical techniques. The authors failed to discuss the theoretical and practical contributions of this study. How did this study provide new knowledge to mining safety and practical recommendations for reducing mining accidents?

Round 2

Reviewer 3 Report

The authors should discuss the limitation of this study that other important explanatory variables such as risk perception have not considered. The authors are suggested to read relevant papers of IJERPH to make the discussion more comprehensive.  
